# Relation of Metal-Binding Property and Selective Toxicity of 8-Hydroxyquinoline Derived Mannich Bases Targeting Multidrug Resistant Cancer Cells

**DOI:** 10.3390/cancers13010154

**Published:** 2021-01-05

**Authors:** Veronika F.S. Pape, Anikó Gaál, István Szatmári, Nóra Kucsma, Norbert Szoboszlai, Christina Streli, Ferenc Fülöp, Éva A. Enyedy, Gergely Szakács

**Affiliations:** 1Institute of Enzymology, Research Centre for Natural Sciences, Hungarian Academy of Sciences, Magyar Tudósok körútja 2, H-1117 Budapest, Hungary; veronika.pape@med.semmelweis-univ.hu (V.F.S.P.); anikgaal@gmail.com (A.G.); kucsma.nora@ttk.mta.hu (N.K.); 2Institute of Chemistry, Eötvös Loránd University, Pázmány Péter sétány 1/A, H-1117 Budapest, Hungary; szobosz@chem.elte.hu; 3Institute of Pharmaceutical Chemistry and Stereochemistry Research Group of Hungarian Academy of Sciences, University of Szeged, Eötvös u. 6, H-6720 Szeged, Hungary or szatmari.istvan@pharm.u-szeged.hu (I.S.); fulop@pharm.u-szeged.hu (F.F.); 4Institute of Atomic and Subatomic Physics, Technical University Vienna, Stadionallee 2, A-1020 Vienna, Austria; streli@ati.ac.at; 5Department of Inorganic and Analytical Chemistry, Interdisciplinary Excellence Centre, University of Szeged, Dóm tér 7, H-6720 Szeged, Hungary; 6MTA-SZTE Lendület Functional Metal Complexes Research Group, University of Szeged, Dóm tér 7, H-6720 Szeged, Hungary; 7Institute of Cancer Research, Medical University of Vienna, Borschkegasse 8a, A-1090 Vienna, Austria

**Keywords:** cancer, multidrug resistance, reactive oxygen species, metal-based drugs, collateral sensitivity

## Abstract

**Simple Summary:**

Effective treatment of cancer is often limited by the resistance of cancer cells to chemotherapy. A well-described mechanism supporting multidrug resistance (MDR) relies on the efflux of toxic drugs from cancer cells, mediated by P-glycoprotein (Pgp). Circumventing Pgp-mediated resistance is expected to make a significant contribution to improved therapy of malignancies. Interestingly, MDR cells exhibit paradoxical hypersensitivity towards a diverse set of anticancer chelators. In this study we explore the relation of chemical and structural properties influencing metal binding and toxicity of a set of 8-hydroxyquinoline derivatives to reveal key characteristics governing “MDR-selective” activity. We find that subtle changes in the stability and redox activity of the biologically relevant metal complexes significantly influence MDR-selective toxicity. Our results underline the importance of chelation in MDR-selective toxicity, suggesting that the collateral sensitivity of MDR cells may be targeted by preferential iron deprivation or the formation of redox-active copper(II) complexes.

**Abstract:**

Resistance to chemotherapeutic agents is a major obstacle in cancer treatment. A recently proposed strategy is to target the collateral sensitivity of multidrug resistant (MDR) cancer. Paradoxically, the toxicity of certain metal chelating agents is increased, rather than decreased, by the function of P-glycoprotein (Pgp), which is known to confer resistance by effluxing chemotherapeutic compounds from cancer cells. We have recently characterized and compared the solution’s chemical properties including ligand protonation and the metal binding properties of a set of structurally related 8-hydroxyquinoline derived Mannich bases. Here we characterize the impact of the solution stability and redox activity of their iron(III) and copper(II) complexes on MDR-selective toxicity. Our results show that the MDR-selective anticancer activity of the studied 8-hydroxyquinoline derived Mannich bases is associated with the iron deprivation of MDR cells and the preferential formation of redox-active copper(II) complexes, which undergo intracellular redox-cycling to induce oxidative stress.

## 1. Introduction

Despite the diversity of drugs used for the treatment of cancer, resistance is a frequent reason for the failure of cancer chemotherapy [1]. Cells that are resistant to a single cytotoxic agent can develop cross-resistance to further structurally and mechanistically unrelated drugs, leading to the phenotype of multidrug resistance (MDR) [2,3]. A well-described mechanism supporting MDR relies on the energy dependent efflux of drugs, resulting in decreased intracellular drug accumulation. Active transport is mediated by ATP-binding-cassette (ABC) proteins, and in particular P-glycoprotein (Pgp, encoded by the ABCB1 gene), which confers resistance to a wide variety of compounds [2,4,5,6,7,8]. Unfortunately, clinical translation of in vitro MDR transporter inhibition proved unsuccessful [7,9,10,11,12,13]. As circumventing Pgp-mediated resistance is expected to make a significant contribution to improved therapy of malignancies, alternative strategies to inhibit, bypass or even exploit efflux-based resistance mechanisms are needed [4,14].

A possible approach to overcome Pgp-mediated MDR exploits cellular vulnerabilities that occur as a result of the adaption to cytotoxic stress. Interestingly, Pgp-expressing MDR cells exhibit paradoxical hypersensitivity towards a diverse set of compounds identified in the Developmental Therapeutics Program of the National Cancer Institute (NCI-DTP) database [15,16,17,18,19,20,21]. In several in vitro models, increased sensitivity of otherwise multidrug resistant cells to the so-called “MDR-selective” compounds was abrogated in the presence of transporter inhibitors, indicating that the activity of Pgp is both necessary and sufficient to confer collateral sensitivity [15,19,20,22]. MDR-selective compounds are enriched in metal chelating ligands [15,19], suggesting that interaction with endogenous metal ions may be fundamental to their mechanism of action. We and others have identified several isatin-*β*-thiosemicarbazones [15,23,24,25], 1,10-phenanthrolines [15,19,26] and 8-hydroxyquinoline derivatives [15,19] with MDR-selective activity, but the molecular determinants of their MDR-selective toxicity have remained elusive. Collateral sensitivity of MDR cells has been associated with preferential ATP depletion resulting from the futile cycling of the Pgp ATPase [16,27,28,29,30,31,32], a differential sensitivity to reactive oxygen species (ROS) [16,18,33], or an increased lysosomal accumulation of compounds [34,35,36]. In view of the metal complex formation ability of the MDR-selective compounds, mechanisms responsible for the toxicity of anticancer chelators deserve special attention. Our earlier work has focused on the role of intracellularly formed complexes in the mechanism of MDR-selective toxicity. In that respect, characterization of solution speciation, the relation of complex equilibria and redox properties to cytotoxic activity, revealed key characteristics governing complex formation and anticancer activity [37,38,39,40,41]. Recently, we have shown that the MDR-selective toxicity of NSC297366 is linked to cellular iron depletion, which is exacerbated by Pgp [42]. Due to their increased proliferation, cancer cells have an excessive demand for iron [43,44] and copper [45,46], and this vulnerability can be exploited by chelator-based cancer treatment strategies [37,39,47,48,49]. The anticancer activity of chelators is explained by the perturbation of the intracellular metal homeostasis, which results either in the depletion of essential metal ions [48,49], or the shuttling of excess metal ions into cellular organelles such as mitochondria [50]. Additionally, metal-binding compounds can influence the activity of various metalloenzymes. Ribonucleotide reductase (RR), which catalyzes the rate limiting step in DNA synthesis, is a well characterized target of anticancer compounds of the thiosemicarbazone class such as Triapine [51,52,53,54]. Anticancer thiosemicarbazones possessing strong metal complex formation abilities were also described as inducing apoptosis by modulating the expression of Bax and Bcl-2 proteins [55,56]. Furthermore, they can cause cell cycle arrest (e.g., by downregulation of Cyclin D1), and inhibit tumor growth through the upregulation of tumor and metastasis suppressors [44,48]. Finally, metal complexes may possess biological activity themselves, partly due to their redox activity [57,58,59].

Interestingly, not every chelator possesses MDR-selective toxicity. Recently, we have characterized the copper(II) and iron(III) binding properties of a set of 8-hydroxyquinoline derived Mannich bases [60]. Since these structurally related analogs possess variable levels of MDR-selective toxicity, our aim here was to understand the relation of chemical and structural properties influencing metal binding to MDR-selective toxicity.

## 2. Materials and Methods

### 2.1. Chemicals

8-Hydroxyquinoline (**Q-1**), 2′,7′-dichlorofluorescein diacetate (DCFDA), CuCl_2_, FeCl_3_, NaOH stock solutions, dimethyl sulfoxide (DMSO), and human holo-transferrin were purchased from Sigma-Aldrich, N-acetyl cysteine (NAC) was purchased from TCI Europe, 3-(4 5-dimethylthiazol-2-yl)-2 5-diphenyltetrazolium bromide (MTT) reagent was obtained from ABCR. Ligands morpholine 7-(morpho-lino-methyl)quinolin-8-ol (**Q-2**) and piperidine 7-(piperidin-1-ylmethyl)quinolin-8-ol (**Q-3**) were obtained from NCI-DTP, 5-chloro-7-((2-fluorobenzylamino)methyl)quinolin-8-ol (**Q-4**) was previously synthesized [60], while **De-Cl-Q-4** without Chloro-substitution in R5 and non-chelating compounds **NC-2** to **NC-4** were synthesized and characterized in this work. Details of the synthetic process and the characterization of the compounds are shown in the supplement. Samples containing the respective ligands and the metal salt (CuCl_2_ or FeCl_3_) at constant metal-to-ligand ratios were prepared as 10 mM stock solutions in an 80%–87% DMSO-water mixture (depending on the metal-to-ligand ratio), by deprotonating the ligand (diluted from a 50 mM to 100 mM DMSO stock) with one equivalent of NaOH and adding the appropriate amount of metal salt stock solutions (100 mM) to obtain the desired metal-to-ligand ratios (1:1, 1:2 or 1:3).

### 2.2. Cell Culture

MDCK II canine kidney cells, A431 epidermoid carcinoma cells, the human uterine sarcoma cell lines MES-SA and the doxorubicin selected MES-SA-Dx5 were obtained from ATCC (MDCK II: No. CRL-2936™, A431: No. CRL-1555™, MES-SA: No. CRL-1976™, MES-SA/Dx5: No. CRL-1977™). ABCB1 was expressed in A431 and MES-SA cells using lentiviral transduction [42,61]. The human cervix carcinoma cell line KB-3-1 and the vinblastine selected KB-v1were kind gifts from Dr. Michael M. Gottesman, National Institutes of Health. The phenotype of the resistant cells was verified using cytotoxicity assays. MDCK-B1 and MDCK-MM were established by the Sleeping Beauty transposon-based gene delivery system [37,62]. OVCAR-8 and NCI-ADRres cells (obtained from the Division of Cancer Treatment and Diagnosis (DCTD) Tumor Repository (National Cancer Institute, Frederick, MD, USA)) were cultivated in RPMI-1640 (Sigma Aldrich, Budapest, Hungary), and other cell lines were cultivated in Dulbecco’s Modified Eagle Medium (DMEM, Sigma Aldrich, Budapest, Hungary), supplemented with 10% fetal bovine serum, 5 mM glutamine, and 50 unit/mL penicillin and streptomycin (Life Technologies, Carlsbad, CA, USA). All cell lines were cultivated at 37 °C, 5% CO_2_.

### 2.3. MTT Viability Assay

MTT viability assays were performed as described earlier with minor modifications [37,63,64]. Briefly, cells were seeded into 96-well tissue culture plates (Sarstedt, Newton, NC, USA/Orange, Braine-l’Alleud, Belgium) at a density of 5000 cells/well and allowed to attach overnight. Test compounds were added to achieve the required final concentration in a final volume of 100 µL per well. After an incubation period of 72 h, the supernatant was removed and fresh medium containing the MTT reagent (0.083 mg/mL) was added. Incubation with MTT at 37 °C was terminated after 1 h by removing the supernatant and lysing the cells with 100 µL DMSO per well. Viability of the cells was measured spectrophotometrically based on the absorbance values at 540 nm using either a Perkin Elmer Victor X3 or an EnSpire microplate reader. Data were background corrected by subtraction of the signal obtained from unstained cell lysates and normalized to untreated cells. Curves were fitted by Prism software (GraphPad Software Inc., San Diego, CA, USA) using the sigmoidal dose-response model (comparing variable and fixed slopes). Curve fit statistics were used to determine the concentration of test compound that resulted in 50% toxicity (IC_50_). Significance was calculated using unpaired *t*-tests; results are given as *: *p* ≤ 0.05, **: *p* ≤ 0.01, ***: *p* ≤ 0.001, ****: *p* ≤ 0.0001.

### 2.4. Total-Reflection X-ray Fluorescence (TXRF) Measurements

Cells were seeded to 6-well plates in a density of 1.5 Mio cells/well and allowed to attach overnight. Following a washing step with phosphate buffered saline (PBS), cells were incubated in serum free DMEM. For the iron-level measurements, cells were preloaded with 25 µM holo-transferrin for 4 h and, after a washing step with serum free DMEM, cells were incubated with 5 µM of each ligand for 8 h. For the detection of copper levels, cells were treated with 1 µM of the ligands in the presence of 5 µM CuCl_2_ for 4 h. Following incubation, cells were harvested upon trypsination, washed twice with PBS, counted and the resulting pellets were digested in 20 μL of 30% H_2_O_2_, 80 μL of 65% HNO_3_. 10 μL of 15 µg/mL Ga(NO_3_)_3_ (in nitric acid) was added as an internal standard upon digestion for 24 h at room temperature. From the resulting solutions, 2 μL were pipetted on the quartz reflectors used for total-reflection X-ray fluorescence (TXRF) analysis. For the determination of the intracellular Cu content, the TXRF method was used as previously reported [65]. Briefly, all determinations were performed on an Atomika 8030C TXRF spectrometer (Atomika Instruments GmbH, Oberschleissheim, Germany). The stock solution of 1000 mg/L Ga was purchased from Merck (Darmstadt, Germany). The Kα line used for determination of Fe and Cu were at 6.403 and 8.047 keV. Applicability of TXRF for the elemental analysis of human cells has been demonstrated earlier [66]. Significance was calculated using unpaired *t*-tests; results are given as *: *p* ≤ 0.05, **: *p* ≤ 0.01, ***: *p* ≤ 0.001, ****: *p* ≤ 0.0001.

### 2.5. Reactive Oxygen Species (ROS) Determination Using DCFDA

Measurements were performed as described earlier [37]. Briefly, cells were harvested, washed with PBS, and incubated with 10 µM DCFDA in a water bath shaker at 37 °C for 30 min in a density of 3 × 10^6^ cells/mL. After washing with PBS, cells were seeded to 96-well plates in PBS in a density of 2 × 10^4^ cells/well. Following the measurement of the basal fluorescence, test compounds were added in different concentrations and the fluorescence of the samples was followed at time intervals of 10 min. DCFDA solution in buffer was used as a cell free control to test for interaction of the test compounds with DCFDA. Data were analyzed as fold change of fluorescence compared to basal levels and untreated cells. Significance was calculated using unpaired *t*-tests; results are given as *: *p* ≤ 0.05, **: *p* ≤ 0.01, ***: *p* ≤ 0.001, ****: *p* ≤ 0.0001.

## 3. Results

In a recent study we have described the synthesis and chemical characterization of four closely related 8-hydroxyquinoline Mannich base derivatives [60]. The series contains structurally related derivatives of the unsubstituted 8-hydroxyquinoline core structure (NSC2039, **Q-1**), including the morpholin-1-yl-methyl derivative NSC662298 (**Q-2**), the piperidin-1-yl-methyl derivative NSC57969 (**Q-3**), and **Q-4**, that was inspired by its ring-closed derivative NSC297366 [19,42] (Appendix A). To assess the effect of these structural modifications on MDR-selective toxicity, herein we included several cell line pairs consisting of drug-sensitive parental and MDR derivatives. Expression of functional Pgp was verified in all MDR cells, which were also characterized for the presence of further ABC-transporters (Appendix A). As shown in Table 1, the compounds possess variable levels of MDR-selective toxicity across the cell panel. **Q-1** is equally toxic to parental MES-SA cells and its MDR derivatives including MES-SA/Dx5 and MES-SA/B1, in which Pgp expression was increased as a result of drug selection [67,68,69] or viral overexpression, respectively. The remaining three ligands exhibited increasing toxicity in the two MDR MES-SA derivatives, while their toxicity remained constant in the Pgp negative MES-SA cells. In particular, **Q-3** and **Q-4** showed marked preferential toxicity in MES-SA/Dx5 and MES-SA/B1 cells, which was abrogated in the presence of the Pgp inhibitor Tariquidar (TQ) (Table 1, TQ data in brackets). The same trends were observed in further cell line pairs including the ovarian cancer cell lines OVCAR-8 and NCI-ADRres, the cervix carcinoma cell lines KB-3-1 and KB-v1, as well as in the epidermoid carcinoma cell line A431 and its transfected counterpart A431-B1. To further confirm the impact of Pgp on the increased toxicity against MDR cells, toxicity was assayed in MDCK cells expressing wild-type Pgp (MDCK-B1) or a non-functional variant (MDCK-MM [62]). Whereas MDCK-B1 cells were more sensitive to **Q-3** and **Q-4**, MDCK cells expressing the inactive Pgp variant did not show collateral sensitivity. These results clearly prove that the increased toxicity of the MDR-selective compounds **Q-3** and **Q-4** is linked to the function of Pgp.

To corroborate the importance of chelation in the MDR-selective anticancer activity of **Q-3** and **Q-4**, we synthesized structural analogues of **Q-2** to **Q-4** without chelating functional groups **NC-2** to **NC-4** by a modified Mannich reaction under microwave conditions (see Appendix A for details of synthesis and characterization, Appendix A). While **Q-1** to **Q-4** are active against the investigated cell lines in the micromolar-to-sub-micromolar concentration range (Table 1 and Table 2), **NC-2** to **NC-4** proved to be significantly (up to two orders of magnitude) less toxic (Table 2). Thus, we conclude that chelation is necessary for MDR-selective toxicity of the investigated compounds, yet not sufficient, as illustrated by the lack of MDR selective toxicity of **Q-1**. While the toxicity of the unsubstituted scaffold **Q-1** is not increased by Pgp, subtle structural modifications resulting in differential toxicity may point to chemical properties that influence MDR-selective activity. Since **Q-4** is chlorinated in position R5, we included derivatives of **Q-1** to **Q-3** with R5-chloro-substitution (**Cl-Q-1** to **Cl-Q-3**) and a **Q-4** derivative without the chloro-substituent (**De-Cl-Q-4**), in order to access the impact of chloro-substitution on observed biological activity (see Appendix A for details of synthesis and characterization, Appendix A). Correlation of the quantifiable chemical properties such as proton dissociation constants (*K*_a_), and solution stability of metal complexes at pH 7.4 [60] with toxicity and selectivity ratios against MES-SA and MES-SA/Dx5 cells are shown in Table 1 and Figure 1A. Chloro-substitution in R5 decreases the p*K*_a_ values of the hydroxyl- as well as of the quinolinium-protons. Analysis of the correlations indicates that a lower p*K*_a_ value of the phenolic OH donor atom and to some degree that of the quinolinium nitrogen is accompanied with increased selective toxicity (Figure 1, compare panels B and C, or D and E, respectively).

### 3.1. Impact of iron(III) and copper(II) Ions on MDR-Selective Cytotoxicity

Similar to p*K*_a_ values, iron(III) and copper(II) binding abilities of the compounds (as reflected in pM* values [60]) also follow the trend of MDR selective anticancer activity (Figure 1A). While for both metal ions a weaker metal binding capacity (at physiological pH) accompanies higher MDR-selectivity, interestingly, in comparison to **Q-1** and **Q-2**, the MDR-selective ligands **Q-3** and **Q-4** show a significantly stronger preference to copper(II) binding over complexation with iron(III) (Figure 1A and Appendix A).

As metal chelation is vital to the toxicity of the investigated 8-hydroxyquinoline derivatives and p*K*_a_ values and metal binding abilities seem to impact the biological activity of the ligands, toxicity studies were performed in the presence of increasing concentrations of iron(III) and copper(II). Strikingly, the four structurally related 8-hydroxyquinoline derivatives showed unique responses to the co-administered metal ions in Pgp positive and parental lines (notably, metal salts alone did not influence cell viability). Whereas co-incubation with iron(III) did not influence the cytotoxicity of the **Q-1** core structure (Figure 2A), it selectively protected MES-SA/Dx5 cells against (**Q-2** and) the MDR-selective derivatives **Q-3** and **Q-4** (Figure 2B–D). The sensitivity of the Pgp negative parental MES-SA cell line towards the investigated derivatives was not significantly affected by co-administration of iron. However, as a result of the selective protection of Pgp positive cells, excess iron ions significantly reduced the MDR-selective toxicity of **Q-3** and **Q-4**. These trends could likewise be observed in the other cell line pairs listed in Table 1 (Appendix A). Derivatives with chloro-substituents in position R5 behaved in a comparable way to their non-chlorinated counterparts (Appendix A).

The experimentally determined stability constants reported in [60] allow the estimation of the concentration distribution of the metal complex species formed at the respective IC_50_ values of the four ligands in the presence of different concentrations of FeCl_3_ or CuCl_2_ (Figure 2E–H). In case of iron, **Q-1** and **Q-2** mostly form tris-ligand complexes (FeL_3_), for **Q-3**, the formation of the bis-complex FeL_2_ is more favored, and the predominant form of **Q-4** is the mono-complex FeL. Interestingly, in the presence of 0.5 µM FeCl_3_, at IC_50_ concentrations of **Q-4** in MES-SA/Dx5 cells a larger fraction of the ligand forms complexes, as compared to the conditions corresponding to IC_50_ values measured in MES-SA cells. These results underline the relevance of iron chelation in the paradoxical sensitivity of MDR cells towards the ligand **Q-4**.

In contrast, co-administration of CuCl_2_ increased the toxicity of **Q-1** and **Q-2** in a dose dependent manner (Figure 2I,J) in both MDR and parental cell lines. Intriguingly, the same cells showed a differential response when copper(II) was combined with MDR-selective derivatives: in the case of ligands **Q-3** and **Q-4**, the parental MES-SA cell line showed a selective increase in sensitivity in the presence of copper, while MDR cells were not further sensitized (Figure 2K,L). As a result, the selective toxicity of **Q-3** and **Q-4** was completely abrogated in the presence of 50 µM copper(II). Again, similar trends were observed in further MDR cells listed in Table 1 (Appendix A). Derivatives with chloro-substituents in position R5 behaved in a comparable way to their non-chlorinated counterparts (Appendix A).

In case of copper(II) co-administration, the type of the formed species strongly depends on the actual CuCl_2_ concentration. Even though at ligand excess (c_L_/c_Cu_ ≥ 2) the coordination sphere is likely to be filled by the available ligands (forming bis-ligand complexes CuL_2_), in the presence of 5 and 50 µM CuCl_2_ (where MDR-selectivity is abrogated) the only species present at IC_50_ concentrations is CuL for all ligands (Figure 2M–P). Strikingly, at lower copper(II) concentrations (where the ligands **Q-3** and **Q-4** still possess MDR selective anticancer activity), the predominant species differ at the respective ligand IC_50_ concentrations in parental MES-SA and MDR MES-SA/Dx5 cells. For example, in the presence of 0.5 µM CuCl_2_, only CuL is predicted to be formed (with a lower fraction of CuLH) at the IC_50_ concentration of **Q-4** in MES-SA/Dx5 cells, while at the IC_50_ against MES-SA, only CuL_2_ and the free ligand are present. Considering the strong preference of MDR-selective ligands to copper(II) ions, these results suggest that the mono-ligand copper complexes might play a decisive role in the selective toxicity of **Q-4**. MES-SA/B1 and MES-SA/Dx5 cells (rendered resistant through overexpressing cDNA-derived Pgp and drug selection, respectively) gave similar results, and their MDR-specific response pattern was abrogated by the Pgp inhibitor tariquidar (TQ) (Figure 3A–D,I–L). Additionally, we repeated the experiments with MDCK cells expressing functional (MDCK-B1) or non-functional Pgp (MDCK-MM) (Figure 3E–H,M–P). Together, these results confirmed the direct involvement of Pgp function in differential sensitivity.

Taken together, co-administration of both iron(III) and copper(II) ions resulted in the selective modulation of MDR-selective toxicity. Interestingly, at constant metal-to-ligand ratios, MDR-selective toxicity of **Q-3** and **Q-4** was retained even in the presence of 0.33, 0.5 or 1.0 equivalents of metal ions (Appendix A), suggesting that for a selective rescue of Pgp positive MES-SA/Dx5 cells by iron(III), as well as for the selective copper(II)-induced sensitization of Pgp negative MES-SA cells, an excess of the respective metal ion is needed. This intriguing result may be explained by an altered metal homeostasis of MDR cells (resulting in, e.g., an increased sensitivity to iron depletion [42]) or a selective induction of downstream toxic effects (such as copper-induced oxidative stress).

### 3.2. Intracellular Metal Levels

To investigate if Pgp-dependent toxicity of **Q-3** and **Q-****4** is linked to a specific effect on metal accumulation, intracellular metal levels were investigated [66,70,71]. Since basal iron levels were at the limit of detection, cells were pre-loaded with 25 µM human holo-transferrin (the iron-saturated protein) prior to treatment with the tested compounds (Figure 4A). In agreement with the selective iron-depleting effect of MDR-selective 8-hydroxyquinoline ligands (*vide supra*, [42]), **Q-4** resulted in a marked, selective iron depletion in MES-SA/B1 cells, which could be abrogated by the Pgp inhibitor TQ. Strikingly, the decrease in intracellular iron levels in MES-SA/B1 cells follows the trend of MDR-selective activity, resulting in a drop of intracellular iron levels below baseline levels in MDR cells treated with **Q-3** or **Q-4**.

Basal intracellular copper levels were much lower than iron levels (Figure 4B). In the presence of 5 µM CuCl_2_
**Q-1**, **Q-2** and **Q-4** significantly increased copper levels in both Pgp negative MES-SA and Pgp positive MES-SA/Dx5 and MES-SA/B1 cells, proportionally to their MDR-selective toxicity **(****Q-1** < **Q-2** < **Q-4,** (Figure 4B)). Copper levels were not consistently different between Pgp negative MES-SA and Pgp positive MES-SA/Dx5 or MES-SA/B1 cells. Thus, selective toxicity towards Pgp positive cells cannot be explained by a mere ligand-induced increase of copper uptake in MDR cells.

### 3.3. Redox Chemistry

Disruption of cellular redox homeostasis can be responsible for the toxicity of an anticancer agent. We previously showed that iron and copper complexes formed with **Q-1** to **Q-4** are redox active [60]. In addition to their redox potentials, redox cycling of the complexed metal ions is also influenced by the kinetics of the reduction and re-oxidation reactions. Reduction of iron(III) and copper(II) complexes by the physiologically relevant reducing agents glutathione (GSH) and ascorbate was followed spectrophotometrically [60]. Although GSH is a stronger reducing agent than ascorbate, ascorbate reacted considerably faster with the iron(III) complexes (Figure 5A), while reduction by GSH seemed to be kinetically hindered (not shown). In case of the copper(II) complexes, however, reduction was observed only with the stronger reducing reagent GSH (Figure 5B). Interestingly, the kinetics of the reduction reactions of in-situ formed iron(III) and copper(II) complexes (with ascorbate and GSH, respectively) were different for the four studied ligands. While in the case of the copper(II) complex formed with **Q-2**, half of the initial complex is reduced by GSH within the first minute of the reaction, for the complex formed with **Q-4**, this state is not yet fully reached after two hours (Figure 5A,B).

Intracellularly formed redox active complexes can lead to the formation of ROS, which can contribute to the toxicity of anticancer compounds. The ability of metal complexes to induce ROS production was investigated using the DCFDA assay. This frequently used reagent is oxidized by ROS to the highly fluorescent 2′,7′-dichlorofluorescein (DCF) upon intracellular cleavage of DCFDA to 2′,7′-dichlorodihydrofluorescein (DCF-H_2_) [72,73]. A fraction of the diacetate ester is also hydrolytically cleaved in the assay buffer solution, thus the oxidation of DCF-H_2_ could also be observed in a cell free condition [37]. In cell-free conditions, the ligands in the presence of equimolar amounts of metal ions (with the exception of the FeCl_3_—**Q-2** mixture) showed only negligible increase in cell-free DCF-fluorescence. However, co-administration of *N*-acetyl cysteine (NAC) resulted in a dose-dependent increase in fluorescence intensity (Figure 5C,D), suggesting that this reducing agent initiated redox cycling by directly reducing the complexed metal ions. As oxidation of the dye is likely due to ROS formed in line with the re-oxidation of the reduced metal ions, this result provides evidence for the redox activity of the formed complexes. Complexes reduced with a higher reaction rate showed a slower re-oxidation rate and vice versa (compare Figure 5A,E/B,F). The kinetics of these reactions can impact the redox equilibria that might interfere with the cellular redox homeostasis [39]. In light of the signals observed in cell-free conditions, experiments using DCFDA in cells should be interpreted with caution.

Incubation of the MES-SA and MES-SA/Dx5 cells with the iron(III) complexes of the investigated ligands resulted in a pronounced increase of DCF emission intensity (Figure 6A), while copper(II) complexes resulted in a smaller initial induction of ROS (Figure 6B). Co-administration of the antioxidant NAC had a paradoxical, dose dependent effect. While a lower concentration of NAC (1 mM) seemed to protect cells against ROS induced by iron and most of the copper complexes, at higher concentrations (5 and 10 mM) DCF fluorescence intensity values were increased. This increase in DCFDA fluorescence is especially pronounced in case of the copper complexes, where an increase in DCF fluorescence could be observed also at the lowest applied NAC concentration in case of the copper complexes of **Q-1** and **Q-4**. Differences between DCF fluorescence observed in Pgp negative and Pgp positive cells were relatively small.

### 3.4. Relevance of Redox Activity of the Complexes on Toxicity—Involvement of ROS

Since iron and copper complexes formed with the reference 8-hydroxyquinoline **Q-1** and its derived Mannich bases **Q-2** to **Q-4** showed redox activity in cell-free systems [60], and ROS was also induced by the complexes in the DCFDA assay (Figure 6), we investigated the effect of the ligands on intracellular ROS production. From the four ligands, **Q-2** showed the most pronounced increase in DCF fluorescence (Figure 7A,B). The signal vanishes in the presence of NAC, suggesting that NAC is scavenging intracellular ROS induced by **Q-2** or its intracellularly formed metal complexes. While for **Q-3** a slight increase can also be observed, this is not the case with **Q-4** (Figure 7A,B). A similar pattern of ROS induction could be observed with the compounds with chloro-substitution in R5 **Cl-Q-1** to **Cl-Q-3** and **De-Cl-Q-4**. (Appendix A). In line with these observations, NAC offered protection against the toxic effect of **Q-2** (and even more against that of **Q-1**) (Figure 7C), suggesting that ROS formation upon intracellular metal complexation might be involved in the mechanism of toxicity of these compounds. Although intracellular redox cycling is feasible for the iron and copper complexes of all four ligands, NAC did not attenuate the toxicity of the MDR-selective derivatives **Q-3** and **Q-4** (Figure 7C), suggesting that **Q-3** and **Q-4** may act in different ways, possibly involving a mechanism of “activation by reduction”. In this model, redox cycling of the metal complexes is initiated by a reducing agent and ROS are formed upon re-oxidation (Figure 8) [37,74].

## 4. Discussion

In our earlier work it has been identified that certain metal chelating 8-hydroxyquinoline derivatives possess robust MDR-selective toxicity, leading to the selective elimination of Pgp-expressing multidrug-resistant cell lines [15,19]. Thus, MDR-selective 8-hydroxyquinoline derived Mannich bases are promising agents for the treatment of drug resistant hyperproliferative diseases, especially for those patients to whom few other therapeutic options remain.

The 8-hydroxyquinoline scaffold is a privileged structure in medicinal chemistry with anticancer, neuroprotective, anti-HIV, but also antifungal, antileishmanial, and anti-schistosomal activities [77,78]. Many of these functions are related to the inhibitory potential of the scaffold against 2-oxoglutarate and iron dependent enzymes, as well as other metalloproteins [77]. A recent report has linked the selective toxicity of di-2-pyridylketone-4,4,-dimethyl-3-thiosemicarbazone (Dp44mT) possessing (N,N,S) metal binding donor set to the Pgp-mediated lysosomal accumulation of a redox-active copper complex [34,35,36]. However, it is unlikely that the structurally unrelated 8-hydroxyquinoline derived Mannich bases would share this mechanism of action, especially considering that increased sensitivity against Dp44mT is restricted to cells with lysosomally localized Pgp [19].

Our earlier work suggested that chelation is a necessary but insufficient prerequisite to elicit MDR-selectivity [19,25]. With the structurally related set of 8-hydroxyquinoline derived Mannich bases possessing increasing MDR-selectivity (**Q-1** to **Q-4**), we extended this observation for this specific compound class by showing that NC-1 and NC-2 without chelating moieties are inefficient. We also demonstrate that the unsubstituted scaffold **Q-1** is not MDR-selective, indicating that chelation per se is not sufficient to convey MDR-selective toxicity. Strikingly, the replacement of a single atom, as realized by the change of the methylene group in the piperidinyl-moiety of **Q-3** to an oxygen atom in the morpholinyl-moiety of **Q-2** resulted in loss of MDR-selective toxicity. In addition to the effect of donor atoms (according to the theory of hard and soft Lewis acids and bases) [79,80], binding preferences of metal ions and the stabilization of respective oxidation states of the metal ions can be fine-tuned by substituents on the scaffold. While 8-hydroxyquinoline (**Q-1**) is a strong chelator with a lower metal binding selectivity, substituted derivatives can exhibit binding preferences [81]. For example, O-Trensox, a water-soluble tripodal ligand based on three 8-hydroxyquinoline subunits selectively binds Fe(III) over divalent metal ions such as copper(II), zinc(II), iron(II) and calcium(II) [81,82]. In this work, we show that that properties influencing metal binding of the 8-hydroxyquinoline derivatives have a strong impact on MDR-selective toxicity. In particular, we find that the p*K*_a_ value of the phenolic OH donor atom is inversely correlated with the MDR-selective toxicity of the compounds (Figure 1). The different p*K*_a_ values, resulting in differently protonated and charged species at physiological pH, may affect the lipophilicity of the ligands. However, the distribution coefficients determined at pH 7.4 (log*D*_7.4_) of the four ligands do not correspond to the observed pattern of MDR-selective toxicity, suggesting that lipophilicity alone is not a predictor of MDR-selective toxicity (see Appendix A). On the other hand, p*K*_a_ values also influence the stability of metal complexes, suggesting that MDR-selective toxicity is affected by metal binding ability [25,79,80,83,84].

Interestingly, the studied MDR-selective compounds show a binding preference for copper(II) over iron(III) (Figure 1C). Copper-chelation by 8-hydroxyquinoline derivatives has been linked to the liberation of ROS and oxidative stress [85]. Iron and copper complexes of the investigated compounds were shown to be redox active [60] under physiological conditions, susceptible to intracellular redox cycling with different kinetics of reduction and re-oxidation reactions (Figure 5A). Formation of intracellular redox active complexes was confirmed with the DCFDA assay. In general, the investigated metal complexes showed only negligible increase in cell-free DCF-fluorescence, but co-administration of NAC resulted in a dose-dependent increase in fluorescence-intensity (Figure 5C,D). This increase of DCF fluorescence is explained by the ability of the antioxidant NAC to initiate redox cycling of the complexes, by directly reducing the chelated metal ions. As shown in Figure 8A, a redox cycle involves reduction and oxidation of the complexed metal ion. Chelating agents modify the redox potential of the free metal ions in favor of either of these two reactions. In the cell, GSH and its precursor NAC usually act as antioxidants [75,76,86], but they can also initiate redox cycling of the complexes by directly reducing the chelated metal ions. Thus, an antioxidant can paradoxically result in an increased formation of H_2_O_2_ or of hydroxyl radicals (OH) via Fenton- and Fenton-like reactions [37,74]. Even though antioxidant treatment did not attenuate the toxicity of the MDR-selective derivatives, NAC-induced increase of DCF fluorescence indicates that ROS can be induced as a result of activation by reduction [37,74], and also suggests that a differential modulation of ROS formation in MDR cells may contribute to selective toxicity.

Recently, we have shown that toxicity of NSC297366 is linked to cellular iron depletion, which is exacerbated by Pgp [42]. In line with this report, co-administration of iron(III) protected Pgp expressing cells from cell death induced by the MDR-selective ligands **Q-3** and **Q-4**. In agreement with a lack of protective effect in the co-administration of FeCl_3_ with **Q-1**, the effect of this ligand on cellular iron levels was found to be negligible (Figure 4A). In contrast, following the order of MDR selective toxicity, **Q-2** to **Q-4** induced a pronounced depletion of iron levels in MES-SA/B1 cells, while their effects on Pgp negative MES-SA cells were comparably low. Underlining the impact of Pgp on this selective effect, inhibition of the transporter by TQ rescued MES-SA/B1 cells from iron depletion, restoring the iron levels to those of the Pgp negative MES-SA cells.

Depending on their lipophilicity, ligands can also induce toxicity through an effect on intracellular copper accumulation [87]. **Q-1**, **Q-2** and **Q-4** significantly increased the uptake of co-administered copper in both Pgp negative MES-SA, and Pgp positive MES-SA/Dx5 and MES-SA/B1 cells, proportionally to MDR-selective toxicity (**Q-1** < **Q-2** < **Q-4**). However, Pgp function did not have an impact on the ligand-induced increase of intracellular copper levels, suggesting that the collateral sensitivity of MDR cells is not simply due to copper deprivation or overload. Interestingly, compared to the other ligands, there was only a modest (10-fold) increase in copper uptake in the presence of **Q-3**. Based on the concentration distribution of complex species previously calculated using the determined stability constants [60], the predominant species in the conditions of this experiment is the mono-ligand complex in all cases. At physiological pH, copper(II)-bound **Q-3** bears a positive charge at the non-coordinating amine of the piperidine moiety. While ligand **Q-4** alone is also protonated at its non-coordinating nitrogen at pH 7.4. However, from the derived stability constants it could be concluded that this moiety is deprotonated when the ligand is coordinated to a copper ion, due to the decreased p*K*_a_ value of the respective complex (L = **Q-4**: log*β*CuLH − log*β* CuL = p*K*_a_ CuLH = 5.91) [60]. As a result, in comparison to complexes formed with ligands **Q-1**, **Q-2** and **Q-4**, those formed with **Q-3** have one additional positive charge. The additional positive charge is expected to hinder cellular uptake, explaining the limited increase of intracellular copper content in case of the **Q-3** complex.

## 5. Conclusions

A promising strategy to overcome multidrug resistance in cancer is to exploit the fitness cost of P-glycoprotein function. In this paper, a set of structurally related 8-hydroxyquinoline derived Mannich-bases with increasing MDR selectivity was investigated with a focus on bioinorganic chemical properties. We show that chelation is key to the mechanism of action of the compounds, suggesting that intracellularly formed metal complexes play an active role in the mechanism of toxicity. Correlation between the p*K*_a_ values, metal binding abilities (reflected as pM* values for iron (III) and copper (II)) and the toxicity of the ligands indicate that subtle changes in the stability and redox activity of the biologically relevant metal complexes may significantly influence MDR-selective toxicity. Co-administration of iron(III) eliminated selective toxicity by protecting MDR cells, indicating that Pgp may increase sensitivity to iron depletion induced by MDR-selective 8-hydroxyquinoline ligands. Future work will be needed to evelaute whether treatment refractory cancer can be clinically targeted by exploiting the fitness cost of MDR. Preliminary studies have shown that physilogical cells expressing Pgp (e.g., brain capillary epithelial cells) do not show collateral sensitivity [19]. However, before clinical trials of MDR-selective compounds can be envisaged, more detailed preclinical studies are needed to determine the best way to deliver these drugs and to establish the proof of concept that MDR-selective compounds can kill transporter-expressing cells in vivo to target transporter-mediated drug resistance [18].

## Figures and Tables

**Figure 1 cancers-13-00154-f001:**
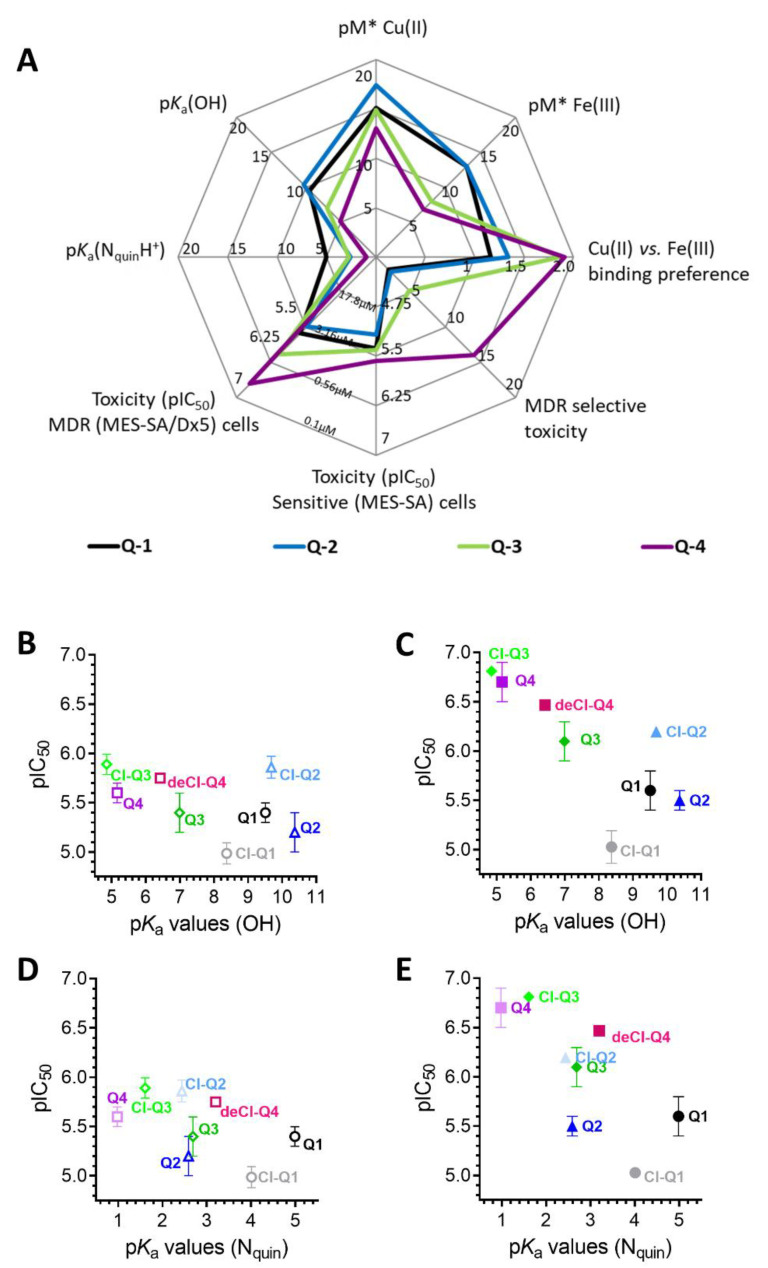
Impact of ligand deprotonation and metal binding properties on MDR-selective anticancer activity. (**A**) Spider diagram showing the relations of solution chemical properties [60] and MDR selective anticancer activity of the four investigated derivatives **Q-1** (black), **Q-2** (blue), **Q-3** (green) and **Q-4** (purple). p*K*_a_ values as well as metal binding capacities of the ligands (expressed as pM* = −log(unbound metal ion)) influence MDR-selective toxicity of 8-hydroxyquinoline derivatives. Detailed correlation plots for the single properties are shown in Appendix A. The impact of p*K*_a_ values of the hydroxyl group (**B**,**C**) and the quinolinium nitrogen (**D**,**E**) on toxicity against MES-SA (**B**,**D**) and MES-SA/Dx5 (**C**,**E**) cells is shown for **Q-1** to **Q-4** as well as for the chlorinated derivatives **Cl-Q-1** (grey), **Cl-Q-2** (light blue), **Cl-Q-3** (bright green), and the **De-Cl-Q-4** (magenta). Toxicity is displayed as pIC_50_ values, the negative decadic logarithm of the half maximal growth-inhibitory concentration.

**Figure 2 cancers-13-00154-f002:**
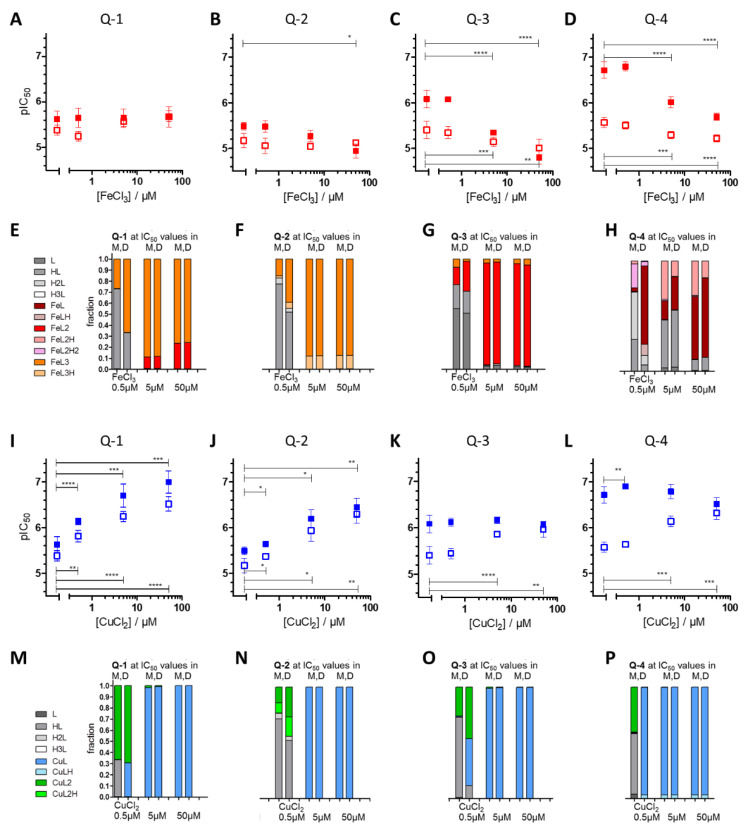
Effect of increasing concentrations of FeCl_3_ (red, **A**–**D**) or CuCl_2_ (blue, **I**–**L**) on the toxicity of the compounds **Q-1** (**A**,**I**), **Q-2** (**B**,**J**), **Q-3** (**C**,**K**) and **Q-4** (**D**,**L**). pIC_50_ values of the compounds were obtained by the MTT assay following 72 h incubation of MES-SA (open squares) and MES-SA/Dx5 (filled squares) cells in the absence or presence of constant metal ion concentrations. Data represent mean values and standard deviations obtained in at least three independent experiments (*: *p* ≤ 0.05, **: *p* ≤ 0.01, ***: *p* ≤ 0.001, ****: *p* ≤ 0.0001). Molar fractions of the complexes formed with ligands **Q-1** to **Q-4** at pH 7.4 at the IC_50_ values of the compounds in MES-SA (M) and MES-SA/Dx5 (D) cell lines upon co-incubation with 0.5, 5, or 50 µM FeCl_3_ (**E**–**H**) and CuCl_2_ (**M**–**P**) were calculated on the basis of experimentally determined stability constants from Reference [60] for **Q-1** (**E**,**M**), **Q-2** (**F**,**N**), **Q-3** (**G**,**O**) and **Q-4** (**H**,**P**). The color code of the single species is given on the left side, and charges are omitted for clarity.

**Figure 3 cancers-13-00154-f003:**
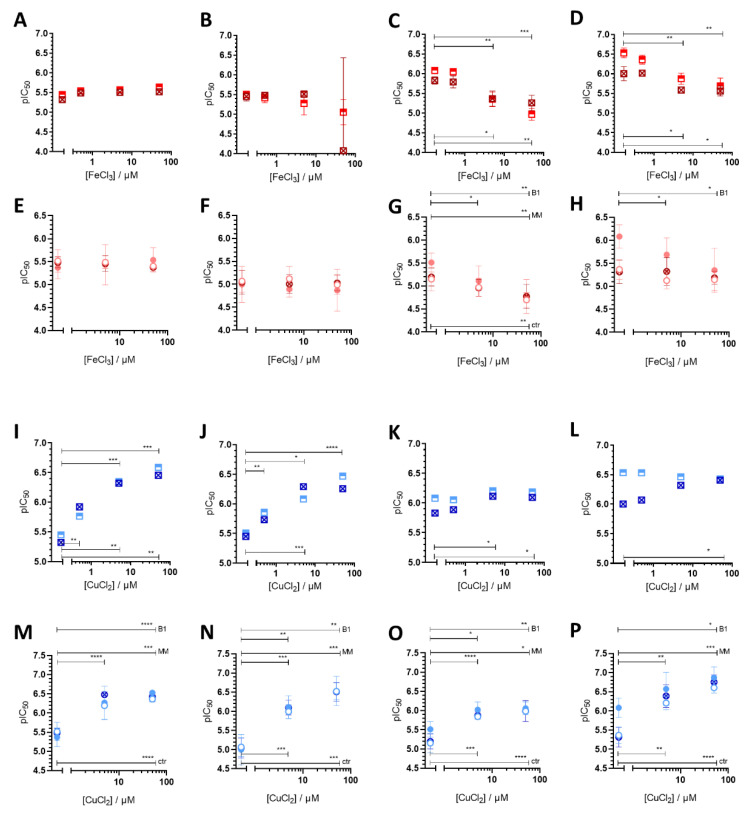
Involvement of Pgp in shaping cellular sensitivity to the ligands **Q-1** (**A**,**E**,**I**,**M**), **Q-2** (**B**,**F**,**J**,**N**), **Q-3** (**C**,**G**,**K**,**O**) and **Q-4** (**D**,**H**,**L**,**P**) in the presence of increasing concentrations of FeCl_3_ (red, **A**–**H**) or CuCl_2_ (blue, **I**–**P**). pIC_50_ values of the ligands in the presence of FeCl_3_ (**A**–**H**) and CuCl_2_ (**I**–**P**) were obtained by MTT assay after 72 h incubation of Pgp expressing MES-SA/B1 cells in the absence (half-filled squares) and in the presence (crossed squares) of the Pgp inhibitor Tariquidar (TQ) (**A**–**D**,**I**–**L**). MDCK cells were used to further confirm the effect of Pgp (Panels **E**–**H** and **M**–**P**). Wild type MDCK cells are displayed with open circles, cells transfected with Pgp (MDCK-B1) with filled circles and MDCK-MM cells transfected with a non-functional mutant MDCK-MM with crossed symbols. Data represent mean values and standard deviations obtained in at least three independent experiments (*: *p* ≤ 0.05, **: *p* ≤ 0.01, ***: *p* ≤ 0.001, ****: *p* ≤ 0.0001).

**Figure 4 cancers-13-00154-f004:**
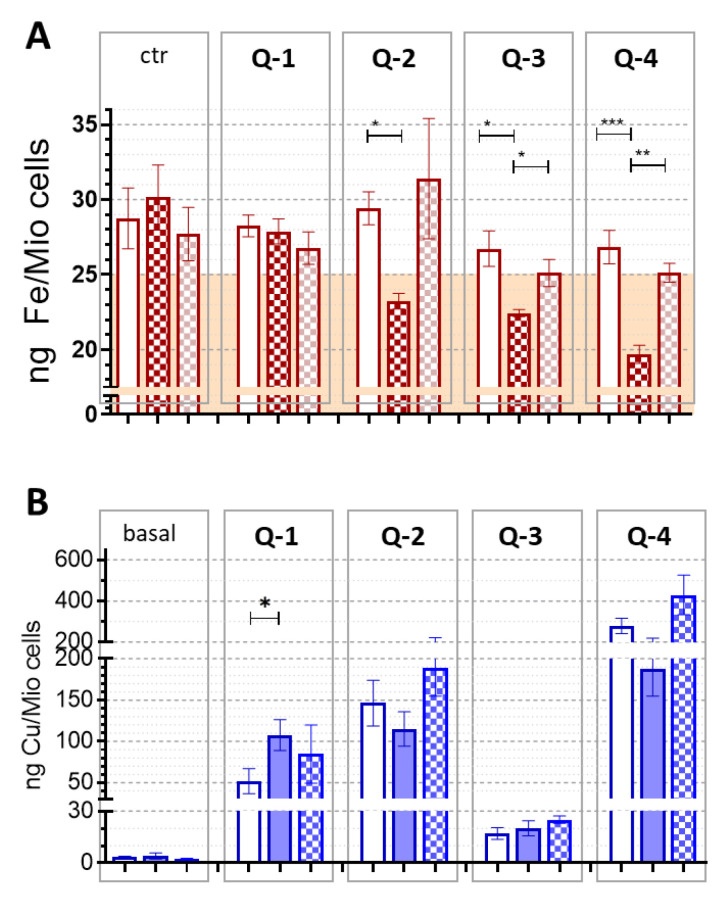
Effect of the compounds on intracellular metal levels. (**A**) Iron levels of MES-SA (open columns) and MES-SA/B1 cells in absence (dark red squared columns) and presence of Pgp inhibitor Tariquidar (bright red squared columns). Cells were loaded with 25 µM holo-transferrin (4 h), followed by an incubation (8 h) without (ctr) or with 5 µM of ligands **Q-1**, **Q-2**, **Q-3**, and **Q-4**. Basal levels (of control cells prior to holo-transferrin-loading) were at a comparable level, indicated by the colored background. (**B**) Cellular copper levels measured following a 4 h co-administration of compounds **Q-1**, **Q-2**, **Q-3**, and **Q-4** (1 µM) and CuCl_2_ (5 µM) in MES-SA (open columns), MES-SA/Dx5 (filled columns) and MES-SA-B1 (squared columns) cells. Significance is given as *: *p* ≤ 0.05, **: *p* ≤ 0.01, ***: *p* ≤ 0.001.

**Figure 5 cancers-13-00154-f005:**
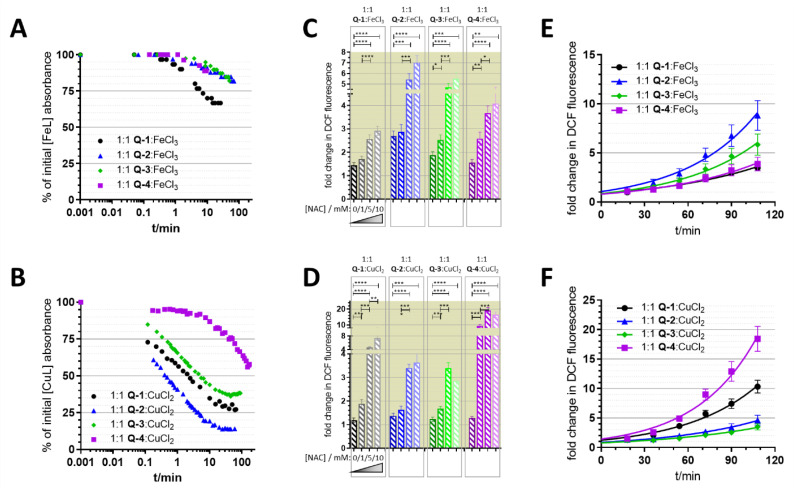
Reduction of the in situ formed iron(III) and copper(II) complexes from equimolar metal-to-ligand ratios with antioxidants was followed spectrophotometrically by the decrease in the respective absorbance maxima. (**A**) Time-dependent decrease of the absorbance values in equimolar 1:1 metal to ligand mixtures of FeCl_3_ with **Q-1** at 356 nm (black), **Q-2** at 348 nm (blue), **Q-3** at 334 nm (green) and **Q-4** at 328 nm upon addition of 2.5 equivalents of ascorbate. (**B**) Time-dependent decrease of the absorbance values in equimolar 1:1 metal to ligand mixtures of CuCl_2_ with **Q-1** at 370 nm (black), **Q-2** at 372 nm (blue), **Q-3** at 368 nm (green) and **Q-4** at 430 nm upon addition of 5 eq GSH. (**C**/**D**): fold changes in DCF-fluorescence (λ_EX_ = 485 nm; λ_EM_ = 535 nm) induced by a 2 h long incubation with 6.25 µM of iron(III) (**C**) and copper(II) (**D**) complexes in the presence of increasing concentrations of N-acetyl cysteine (NAC) (indicated by fading color from 0 to 1, 5, 10 mM). Results of complexes of **Q-1** are shown in black, of **Q-2** in blue, **Q-3** in green and **Q-4** in purple. (**E**,**F**) Kinetics of the DCFDA-oxidation reactions in the presence of 10 mM NAC from panels (**C**,**D**). *: *p* ≤ 0.05, **: *p* ≤ 0.01, ***: *p* ≤ 0.001. ****: *p* ≤ 0.0001.

**Figure 6 cancers-13-00154-f006:**
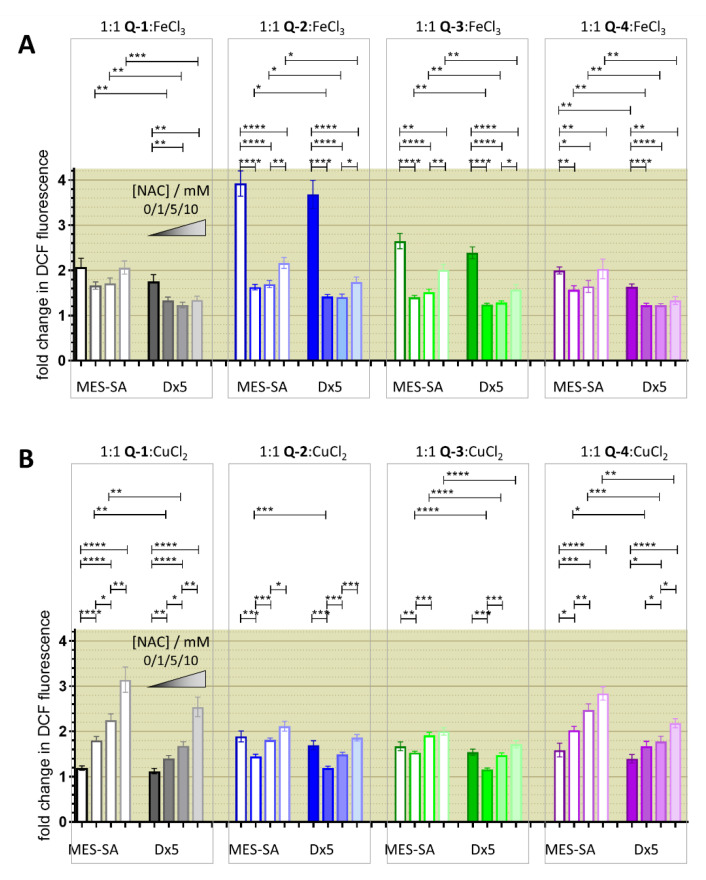
Characterization of intracellular ROS production with the DCFDA assay. Fold changes in fluorescence are shown after a 90 min incubation with 6.25 µM of the ligands **Q-1** to **Q-4** in the presence of an equimolar amount of iron (**A**) and copper (**B**). Values are given as mean with SEM of three-to-five independent experiments performed in duplicate. NAC increases the DCF fluorescence upon treatment in Pgp negative MES-SA (open columns) and Pgp positive MES-SA/Dx5 (Dx5, filled columns) in a concentration dependent manner (fading color indicates increasing NAC concentration from 0, 1, 5, 10 mM). Significance is given as *: *p* ≤ 0.05, **: *p* ≤ 0.01, ***: *p* ≤ 0.001, ****: *p* ≤ 0.0001.

**Figure 7 cancers-13-00154-f007:**
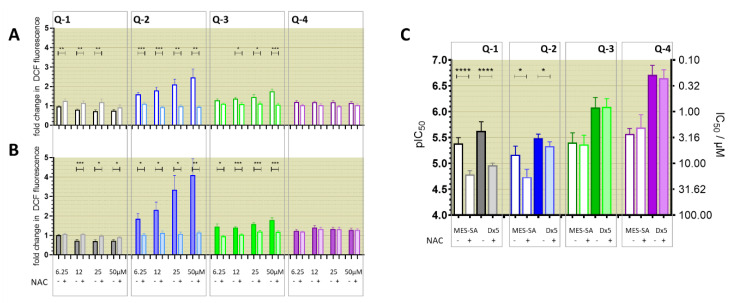
Intracellular ROS production was measured upon incubation of MES-SA (**A**) and MES-SA/Dx5 (**B**) cells with increasing concentrations of **Q-1** to **Q-4** ranging from 6.25 µM to 50 µM in the absence (darker) or presence (lighter) of 5 mM NAC. Fold changes in fluorescence are shown after a 2 h incubation with the respective compounds. Data of the differently chlorinated ligands as well as of cell free controls are shown in Appendix A. (**C**) Effect of NAC (5 mM) on the toxicity of **Q-1** to **Q-4**. Average pIC_50_ values and standard deviations are shown for Pgp-negative MES-SA (open columns) and Pgp-positive MES-SA/Dx5 cells (filled columns) in the absence (intense) and presence (pale) of 5 mM NAC. For comparison, the respective µM concentrations are shown on the right *y*-axis. NAC alone (5 mM) did not influence the viability of the cells (see Appendix A). Significance was calculated using unpaired *t*-tests; results are given as *: *p* ≤ 0.05, **: *p* ≤ 0.01, ***: *p* ≤ 0.001, ****: *p* ≤ 0.0001.

**Figure 8 cancers-13-00154-f008:**
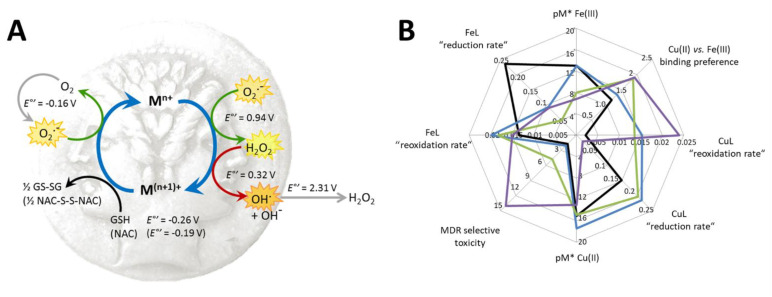
(**A**) Redox cycling metal ions (e.g., as part of redox active complexes, blue arrows), enable the formation of ROS (superoxide O_2_^−^, hydrogen peroxide H_2_O_2_, hydroxyl radical OH). Formal redox potentials of respective ROS are given in 1 M solutions at physiological pH [58]. Antioxidants are Janus-faced: based on their formal redox potentials, GSH and NAC can scavenge ROS [75,76], but they can also reduce the metal ion (black arrow), thus inducing a mechanism of “activation by reduction” [37,74], giving rise to Fenton (iron)/Fenton-like (other redox active metal ions) reactions (red arrow). In the metalloenzyme superoxide dismutase (SOD), the redox active copper center catalyzes the reactions highlighted by green arrows. (**B**) Summary of metal binding preferences and redox kinetic parameters of **Q-1** (black), **Q-2** (blue), **Q-3** (green) and **Q-4** (purple). pM*: metal binding capacities of the ligands = −log(unbound metal ion).

**Table 1 cancers-13-00154-t001:** Toxicity of the studied compounds in a panel of multidrug resistant (MDR) cell lines. IC_50_ values (50% toxicity) are shown in µM, as determined by MTT assays performed in the absence or presence of 1 µM Tariquidar (values in brackets). MDR-selective toxicity of a compound is expressed as the fraction of IC_50_ values obtained in Pgp negative vs. positive cells (selectivity ratio, SR) determined from 3–18 independent experiments. A compound is considered to possess MDR-selectivity at SR > 2. Significance was calculated using unpaired *t*-tests; results are given as *: *p* ≤ 0.05, **: *p* ≤ 0.01, ***: *p* ≤ 0.001, ****: *p* ≤ 0.0001.

IC_50_ (µM)	Q-1	SR	Q-2	SR	Q-3	SR	Q-4	SR
MES-SA	4.19 ± 0.74 (5.08 ± 0.48)		6.97 ± 1.75 (4.65 ± 0.92)		4.11 ± 1.20 (2.99 ± 0.91)		2.75 ± 0.46 (2.22 ± 0.36)	
MES-SA/Dx5	2.46 ± 0.69 ^a^ (2.94 ± 0.66)	1.70 ** (1.73 *)	3.27 ± 0.39 ^a^ (5.49 ± 1.02)	2.13 *** (0.85)	0.86 ± 0.26 ^a^ (3.15 ± 0.93)	4.75 **** (0.95)	0.20 ± 0.06 ^a^ (3.28 ± 0.51)	13.68 **** (0.68)
MES-SA/B1	3.68 ± 0.95 (4.92 ± 1.32)	1.14 (1.03)	3.12 ± 0.28 (3.60 ± 0.63)	2.23 *** (1.29)	0.83 ± 0.02 (1.50 ± 0.22)	4.95 **** (1.99)	0.30 ± 0.06 (1.03 ± 0.29)	9.24 **** (2.16)
OVCAR-8	3.45 ± 0.13 (3.43 ± 0.40)		3.16 ± 0.52 (3.28 ± 0.41)		3.15 ± 0.36 (3.72 ± 0.55)		1.08 ± 0.10 (1.09 ± 0.12)	
NCI-ADRres	1.84 ± 0.30 (2.65 ± 0.8)	1.88 (1.29)	2.84 ± 0.81 (5.41 ± 1.55)	1.11 (0.61)	0.68 ± 0.09 (4.22 ± 0.86)	4.60 ** (0.88)	0.14 ± 0.07 (1.15 ± 1.24)	7.84 * (0-95)
KB-3-1	8.59 ± 2.92 (10.89 ± 3.04)		9.55 ± 2.55 (11.69 ± 3.31)		6.19 ± 0.12 (6.69 ± 0.15)		1.75 ± 0.62 (2.65 ± 1.57)	
KB-v1	8.71 ± 4.43 (10.85 ± 2.52)	0.99 (1.00)	6.42 ± 2.19 (11.34 ± 3.11)	1.49 (1.03)	1.56 ± 0.57 (7.47 ± 0.84)	3.96 * (0.89)	0.27 ± 0.12 (2.27 ± 0.43)	6.51 * (1.17)
A431	2.35 ± 0.92 (2.90 ± 1.37)		3.31 ± 1.61 (3.18 ± 1.45)		3.91 ± 0.83 (4.09 ± 0.88)		5.82 ± 1.15 (4.54 ± 1.42)	
A431-B1	8.21 ± 4.04 (7.06 ± 4.75)	0.31 (0.41)	4.05 ± 1.86 (7.64 ± 5.32)	0.82 (0.42)	1.14 ± 0.16 (5.58 ± 1.67)	3.96 *** (0.73)	2.57 ± 0.56 (7.95 ± 2.62)	2.26 *** 0.57*)
MDCK	3.27 ± 1.24 (3.08 ± 1.41)		9.15 ± 3.51 (11.10 ± 4.87)		7.61 ± 3.06 (10.18 ± 3.12)		4.55 ± 1.53 (5.92 ± 2.01)	
MDCK-B1	4.58 ± 1.75 (15.17 ± 10.11)	0.71 (0.20)	11.98 ± 7.84 (12.08 ± 5.07)	0.76 (0.92)	3.22 ± 1.02 (12.41 ± 6.27)	2.36 * (0.82)	0.89 ± 0.35 (6.95 ± 3.49)	5.13 **** (0.85)
MDCK-MM	3.34 ± 0.60 (3.59 ± 1.01)	0.97 (0.86)	9.74 ± 4.04 (9.01 ± 3.80)	0.82 (1.23)	6.67 ± 2.00 (8.45 ± 2.30)	1.14(1.20)	5.23 ± 2.11 (6.27 ± 2.29)	0.87 (0.94)

^a^ Data are taken from Ref. [60].

**Table 2 cancers-13-00154-t002:** pK_a_ values and chemical structures of **Q-1** to **Q-4** and related compounds determined by UV-visible spectrophotometric titrations (T = 25 °C, I = 0.2 M (KCl)). The non-chelating derivatives 2-(morpholino-methyl)naphthalen-1-ol **NC-2**, 2-(piperidin-1-ylmethyl)naphthalen-1-ol **NC-3** and 2-((2-fluorobenzylamino)methyl)naphthalen-1-ol **NC-4** were used to monitor the lack of the quinoline nitrogen. Significance was calculated using unpaired *t*-tests; results are given as *: *p* ≤ 0.05, **: *p* ≤ 0.01, ***: *p* ≤ 0.001, ****: *p* ≤ 0.0001.

Compound	Structure	MES-SA	MES-SA/Dx5	SR_MES-SA/Dx5_	p*K*_a_(OH)	p*K*_a_(N_quinolinium_H^+^)	p*K*_a_(additional)
**Q-1**	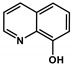	4.19 ± 0.74(5.08 ± 0.48)	2.46 ± 0.69 ^a^(2.94 ± 0.66)	1.70 **(1.73 *)	9.51 ^a^	4.99 ^a^	
**5-Cl-Q-1**	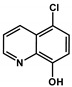	10.44 ± 1.71(12.10 ± 0.42)	9.66 ± 2.48(8.32 ± 0.53)	1.08(1.46 *)	8.37 ^b^	4.01 ^b^	
**Q-2**	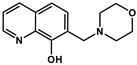	6.97 ± 1.75(4.65 ± 0.92)	3.27 ± 0.39 ^a^(5.49 ± 1.02)	2.13 ***(0.85)	10.37 ^a^	2.59 ^a^	6.25 ^a^(N_morpholinium_H^+^)
**5-Cl-Q-2**	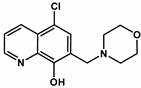	1.40 ± 0.24(1.57 ± 0.17)	0.64 ± 0.03(1.13 ± 0.28)	2.19 *(1.39)	9.68 ± 0.03	<2	5.83 ± 0.02 (N_morpholinium_H^+^)
**NC-2**	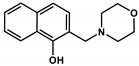	421.68 ± 34.90	561.61 ± 36.8	0.75	9.21 ± 0.02 ^c^	‒	6.61 ± 0.02 ^c^ (N_morpholinium_H^+^)
**Q-3**	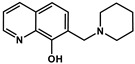	4.11 ± 1.20(2.99 ± 0.91)	0.86 ± 0.26 ^a^(3.15 ± 0.93)	4.75 ****(0.95)	6.99 ^a^	2.69 ^a^	>11.5 ^a^
**5-Cl-Q-3**	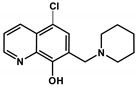	1.30 ± 0.21(1.46 ± 0.15)	0.15 ± 0.01(0.90 ± 0.12)	8.40 ***(1.61 *)	4.85 ± 0.02	1.61 ± 0.04	7.16 ± 0.03(N_piperidinium_H^+^)
**NC-3**	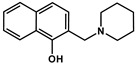	139.53 ± 47.89	162.78 ± 16.63	0.86	6.06 ± 0.01 ^c^	‒	10.12 ± 0.01 ^c^(N_piperidinium_H^+^)
**De-Cl-Q-4**	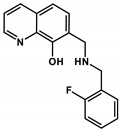	1.78 ± 0.08(2.25 ± 0.54)	0.34 ± 0.01(1.31 ± 0.04)	5.22 **(1.72)	6.42 ± 0.04	3.21 ± 0.04	10.69 ± 0.04 (N_benzylamine_H_2_^+^)
**Q-4**	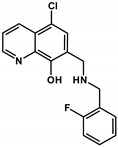	2.75 ± 0.46(2.22 ± 0.36)	0.20 ± 0.06 ^a^(3.28 ± 0.51)	13.68 ****(0.68)	5.16 ^a^	< 2 ^a^	8.54 ^a^(N_benzylamine_H_2_^+^)
**NC-4**	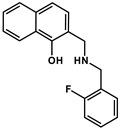	44.69 ± 4.52	31.09 ± 2.17	1.44	5.50 ± 0.02	‒	9.72 ± 0.02 (N_benzylamine_H_2_^+^)

^a^ Data are taken from Ref. [58]. ^b^ Predicted by the MarvinSketch software (ChemAxon Ltd. Budapest, Hungary). Notably, **5-Cl-Q-1** has extremely bad water solubility hindering the accurate determination of its values. ^c^ Due to the overlapping deprotonation processes the assignation of the p*K*_a_ values to the various moieties is uncertain.

## Data Availability

The data presented in this study are contained within the article and the Appendix A.

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
