# Peer review of "Relation of Metal-Binding Property and Selective Toxicity of 8-Hydroxyquinoline Derived Mannich Bases Targeting Multidrug Resistant Cancer Cells"

_cancers, 2021, doi:10.3390/cancers13010154_

Round 1
Reviewer 1 Report
The authors have carried out several complementary experiments and answered most of the initial reviewers comments. This greatly improved the overall quality of their initial submission, which should be suitable for publication in Cancers after only minor revisions, concerning a point raised by reviewer 2 (see below).
Revision requested:
Reviewer 2 statement :
Many experiments seem to have been carried out using the Cu or Fe complex rather than the ligand (see Figure 5 for example). Yet, the preparation or the structure of the complexes is not showed (one has to refer to ref 58, which only includes the structure of the Cu complexes). This should be included.
The authors replied :
« Unfortunately, we were not able to isolate crystals of the iron complexes for X-ray analysis. However, complexation of iron with 8OHQ ligands is well known in the literature. As we showed in Ref 58, the investigated ligands coordinate to copper in the expected way (via bidentate N,O- binding mode). Hence, a similar coordination mode can be expected for the iron complexes.
The focus of this study was to link metal binding properties to MDR-selective activities of the investigated ligands, rather than the solid state characterization of the complexes. »
The focus of the present study is certainly the biological activity and not the structural characterization, yet this does not precludes including the protocol for the synthesis of the complexes. The procedure for the synthesis, isolation and characterization of the complexes must be added in the supplementary information. For Figure 5 the authors state that a mono complex (ML) is used, but in ref 58 ML and ML2 complexes are described. The isolation of crystals suitable for X-ray characterization is indeed not needed, but the preparation of the complexes, indicating the ligand/metal ratio, must be included. Without this information the data from figures 5 and 6 are not entirely conclusive and cannot be reproduced.
Author Response
We thank the reviewer for this comment. Indeed, the legend to Figures 5 and 6 suggested that we isolated the respective complexes whereas these were prepared in situ. This is now clearly stated in the modified figures to indicate that pre-mixed solutions were added at a given metal-to-ligand ratio. Further, the preparation of the samples containing the ligands and the metal salts is described in the Chemicals section:
“Samples containing the respective ligands and the metal salt (CuCl2 or FeCl3) at constant metal-to-ligand ratios were prepared as 10 mM stock solutions in a DMSO-water mixture, by deprotonating the ligand (diluted from a 50 mM to 100 mM DMSO stock) with one equivalent of NaOH and adding the appropriate amount of metal salt stock solutions (100 mM) to obtain the desired metal-to-ligand ratios (1:1, 1:2 or 1:3).”
Reviewer 2 Report
Very interesting point of studying the relation of iron and copper binding property of 8-hydroxyquinoline derivatives.
The introduction provides sufficient background data, but I recommend to focus more on their work, since the authors have valuable contributions to the field.
It is clear how this woks in vitro but not clear how this could be translated into preclinical studies in order to elucidate unknown mechanisms, I recommend to add a paragraph of perspectives.
Author Response
We thank the Reviewer for the positive comments. As requested, we added further references to our earlier work focusing on the MDR-selective toxicity of anticancer chelators (refs 38-42), and we included a short concluding paragraph on the prospect of the preclinical development of these compounds.
Reviewer 3 Report
In this manuscript, the authors investigated the collateral sensitivity of 8-hydroxyquinoline and its Mannich base derivatives. Q3 and Q4 showed better collateral sensitivity towards Pgp overexpressing cancer cells, accompanied with stronger chelating effects with Cu over Fe. The authors also studied the influence of different variation of structures, which may provide useful information for the development of this type of anticancer agent. While the results are interesting and show potential collateral sensitivity to Pgp-mediated MDR cancers, the presentation is obscure and some results are confusing and paradoxical. Overall, the manuscript needs major revision with better interpretation of some results. The manuscript may be better fitted for a computational chemistry or medicinal chemistry journal.
Major
- It is unclear what structure is the key pharmacophore. Is it the original Q3 and Q4, or the coordinators composed of Q3/Q4-chelated Fe/Cu?
- The statement in lines 331-332 is problematic. In Figure 5A,B, the authors tested/compared the kinetic behavior of coculture of Fe with Q1-Q4 in the presence of VitC with that of Cu with Q1-Q4 in the presence of GSH. Because two variables, Fe, Vit C vs Cu, GSH were introduced in each group, it is unclear how the conclusion was derived.
- The findings in Figure 7 suggest that the most important Q3 and Q4 did not induce cell death or collateral sensitivity via ROS. Yet, no other mechanism has been proposed/investigated.
- More studies should be performed to compare Q3 and Q4 with similar structure and inducing collateral sensitivity but exhibiting quite different Cu accumulation (chelation) and ROS induction.
- Discussions on limitations of this study including potential toxicity should be provided. The mechanisms/connections of these chelators to Pgp are not addressed and discussed.
- The connection among chemical/structural properties, metal binding and collateral sensitivity is not clearly described. Furthermore, the findings are inconsistent. For example, it was concluded that Q-1 is not MDR-selective although Table 1 showed certain MDR-selective property. It is also unclear what Cl- does to chelation and MDR-selective property although the replacement of N with C may lead to the loss of chelation and total loss of potency.
Minor
- A Figure showing all the structures of 8-hydroxyquinoline Mannich base derivatives tested in the study should be included for better understanding and comparison of the chemical structures/properties that underlie the mechanism of metal chelating.
- All cells used should be included in Section 2.2
- Statistics method should be included in the method section.
- The asterisks * (**, ***) in Table 2 are not explained.
- Line 153, MDR has already been defined in the Introduction.
- de-Cl-Q-4 or 5-De-Cl-Q-4?
- Define pIC50
- Methods should be provided to clearly indicate whether the results shown in Figs 2E-H, M-P are predicted or experimental.
- Statistical differences should be analyzed in Figures 4 and 7.
Author Response
Comments and Suggestions for Authors
In this manuscript, the authors investigated the collateral sensitivity of 8-hydroxyquinoline and its Mannich base derivatives. Q3 and Q4 showed better collateral sensitivity towards Pgp overexpressing cancer cells, accompanied with stronger chelating effects with Cu over Fe. The authors also studied the influence of different variation of structures, which may provide useful information for the development of this type of anticancer agent. While the results are interesting and show potential collateral sensitivity to Pgp-mediated MDR cancers, the presentation is obscure and some results are confusing and paradoxical. Overall, the manuscript needs major revision with better interpretation of some results. The manuscript may be better fitted for a computational chemistry or medicinal chemistry journal.
Major
- It is unclear what structure is the key pharmacophore. Is it the original Q3 and Q4, or the coordinators composed of Q3/Q4-chelated Fe/Cu?
The goal of this study was to characterize the impact of iron(III) and copper(II) on the MDR-selective toxicity of structurally related 8-hydroxyquinoline derived Mannich bases. Clearly, the ligands (without the addition of metals) possess MDR-selective anticancer activity (Tables 1, 2), suggesting that the 8-hydroxyquinoline derived Mannich base scaffold (including the methylene linked amine) may be considered as a “pharmacophore”. However, our results also show that chelation is key to the mechanism of action of the compounds (see Table 2). Therefore, we suspect that intracellularly formed complexes play an active role in the mechanism of toxicity. The anticancer activity of chelators is usually explained by the perturbation of the intracellular metal homeostasis, which results either in the depletion of essential metal ions or the shuttling of excess metal ions into the cells. Several pathways are affected, but the exact molecular targets of (MDR-selective) anticancer chelators are not known. For these reasons, the pharmacophore cannot be clearly defined at present. Work in our laboratory is ongoing to synthesize focused libraries, with the aim to identify those chemical features that are key to the MDR-selectivity of the compounds.
- The statement in lines 331-332 is problematic. In Figure 5A,B, the authors tested/compared the kinetic behavior of coculture of Fe with Q1-Q4 in the presence of VitC with that of Cu with Q1-Q4 in the presence of GSH. Because two variables, Fe, Vit C vs Cu, GSH were introduced in each group, it is unclear how the conclusion was derived.
We thank the Reviewer for this comment. Although GSH is a stronger reducing agent than ascorbate, the latter compound was able to reduce iron(III) complexes, but we did not observe measurable reduction of the iron(III) complexes by GSH (probably due to kinetic control of this reaction). In contrast, copper(II) complexes were reduced by GSH, but not by the weaker reducing agent ascorbate. For these reasons, we were not able to directly compare iron and copper reducibility. Instead, we show that within one system (iron – ligand – ascorbate, or copper – ligand – GSH) the complexes formed with the respective ligands behave quite differently (Figure 5 A and B). If we look at the kinetics of the increase of emission intensity of DCF as an indicator for re-oxidation of the complex, then, interestingly, the complex characterized by the slowest reduction, is oxidized the fastest and vice versa. Hence, we conclude that redox kinetics might play an important role.
We added a detailed explanation of the experiment to clarify this point.
- The findings in Figure 7 suggest that the most important Q3 and Q4 did not induce cell death or collateral sensitivity via ROS. Yet, no other mechanism has been proposed/investigated.
Indeed, findings in Figure 7 suggest, that ROS induction by the ligands Q-3 and Q-4 is less pronounced than in case of Q-2. However, in the presence of metal ions, the concentration dependent NAC-induced effect of ligands on DCF fluorescence (Figure 6) indicates that ROS can be induced by means of “activation by reduction”.
At this point we can only speculate about the mechanisms underlying selective toxicity. Recently, we have shown that NSC297366 induces iron depletion, which is exacerbated by Pgp (ref. 38 – now 43). In line with this report, co-administration of iron(III) protected Pgp expressing cells from cell death induced by the MDR-selective ligands Q-3 and Q-4. In agreement with a lack of protective effect in the co-administration of FeCl3 with Q-1, the effect of this ligand on cellular iron levels was found to be negligible (Figure 4 A). In contrast, following the order of MDR selective toxicity, Q-2 to Q-4 induced a pronounced depletion of iron levels in MES-SA/B1 cells, while their effects on Pgp negative MES-SA cells were comparably low. Underlining the impact of Pgp on this selective effect, inhibition of the transporter by TQ rescued MES-SA/B1 cells from iron depletion, restoring the iron levels to those of the Pgp negative MES-SA cells.
- More studies should be performed to compare Q3 and Q4 with similar structure and inducing collateral sensitivity but exhibiting quite different Cu accumulation (chelation) and ROS induction.
Our earlier results established the 8-hydroxyquinoline Mannich base backbone as a promising scaffold for designing MDR-targeting compounds. Using Q-3 and Q-4 as starting points, we are currently exploring further 8-hydroxyquinoline derivatives to establish structure–activity relationships governing selective toxicity. Unfortunately, prediction of the complex formation constants is notoriously difficult, as various substituents have a strong influence on the stability and stoichiometry of the complexes. Laborious experiments, beyond the scope of the revision process, will be needed to fully explore the link between Cu(II) complex formation/accumulation, ROS induction, and MDR-selective toxicity.
- Discussions on limitations of this study including potential toxicity should be provided. The mechanisms/connections of these chelators to Pgp are not addressed and discussed.
We added a concluding paragraph that addresses the limitation of our conclusions, highlighting challenges associated with preclinical development (see also answers to Reviewer 2).
There are several lines of evidence linking the toxicity of these chelators to Pgp. First, cells engineered to overexpress Pgp by lentiviral transfection (MES-SA/B1, A431-B1, MDCK-B1) become sensitive. Second, increased toxicity is abrogated in the presence of a specific Pgp inhibitor. Third, overexpression of a non-functional Pgp variant in MDCKII-MM cells does not induce collateral sensitivity. Taken together, these results clearly show that the toxicity of the MDR-selective 8-hydroxyquinoline derivatives is mediated by the activity of Pgp. Potential mechanisms are addressed in the Introduction and the Discussion.
- The connection among chemical/structural properties, metal binding and collateral sensitivity is not clearly described. Furthermore, the findings are inconsistent. For example, it was concluded that Q-1 is not MDR-selective although Table 1 showed certain MDR-selective property. It is also unclear what Cl- does to chelation and MDR-selective property although the replacement of N with C may lead to the loss of chelation and total loss of potency.
Correlation of the quantifiable chemical properties with the toxicity and selectivity ratios against parental MES-SA and MDR MES-SA/Dx5 cells are shown in Table 1 and Figure 1A. The data indicate that a chloro substitution in R5 decreases the pKa values of the hydroxyl- as well as of the quinolinium-nitrogen-protons. As shown in Figure 1, panels B-E, MDR-selective activity seems to correlate with the pKa values of the phenolic-OH and quinolinium nitrogen of the investigated 8-hydroxyquinoline -derived Mannich bases. The stabilities of the iron(III) or copper(II) complexes of the differently chlorinated derivatives were not determined (due to their lower water solubility). Chloro-substitution in R5 seems to increase MDR-selective anticancer activity. In parallel, the pKa values of the chlorinated derivatives are lower than those of their non-chlorinated counterparts; the newly added compounds with different chlorination pattern (Cl-Q-1, Cl-Q-2, Cl-Q-3 and De-Cl-Q4) follow the trend that was suggested by the four originally investigated derivatives. Taken together, the data suggest that a lower pKa value of the phenolic OH donor atom and to some degree also that of the quinolinium nitrogen is accompanied with increased selective toxicity.
Note that selective toxicity is defined as SR>2, meaning an at least 2-fold sensitivity of MDR cells (this is now clearly stated in the revised text. Thus, although Q1 is slightly more toxic to certain MDR cells, it is not classified as an MDR-selective compound.
Minor
- A Figure showing all the structures of 8-hydroxyquinoline Mannich base derivatives tested in the study should be included for better understanding and comparison of the chemical structures/properties that underlie the mechanism of metal chelating.
A new supplementary figure (Figure S1) was added to the text.
- All cells used should be included in Section 2.2
Section 2.2 was updated with the cell line references.
- Statistics method should be included in the method section.
- The asterisks * (**, ***) in Table 2 are not explained.
The statistics methods are now included in the Methods, the asterisks are explained.
- Line 153, MDR has already been defined in the Introduction.
- de-Cl-Q-4 or 5-De-Cl-Q-4?
- Define pIC50
Corrected, as requested
- Methods should be provided to clearly indicate whether the results shown in Figs 2E-H, M-P are predicted or experimental.
The figure legend was revised (the molar fractions are predicted/calculated based on experimentally determined (published) stability data.
- Statistical differences should be analyzed in Figures 4 and 7.
The missing statistics were added to Figures 4 and 7.
Round 2
Reviewer 1 Report
The explanation about the preparation of the metal complexes that have been evaluated is now satisfying and clear for the readers. In this present form this manuscript should be publishable in Cancers.
In the rewritten conclusion there are few typos that need to be corrected:
"evalaute" / "physilogical"
This manuscript is a resubmission of an earlier submission. The following is a list of the peer review reports and author responses from that submission.
Round 1
Reviewer 1 Report
The manuscript by Pape et al. reports the effect of hydroxyquinoline derived Mannich bases in term of toxicity, redox activity and oxidative stress of their iron and copper complexes on MDR-cancer cell lines. Different levels of MDR-mediated toxicity are reported depending on metal binding properties of 8-hydroxyquinoline compounds and correlated with chemical/structural properties likely influenced by metal binding.
Overall the paper is clearly written and technically sounds, and evidence is supported by the experimental data.
The expression levels of PgP in the different cell lines upon drug exposure could be added for completeness of data.
Reviewer 2 Report
General assessment
This manuscript by Pape et al. describes the thorough evaluation of the metal-binding properties, redox activities and other physico chemical parameters of four 8-hydroxyquinoline derivatives with regard to their selective toxicity against MDR cancer cells. This is follow-up to their previous studies on these four molecules (ref 15, 19, 58).
Overall, the paper is well written and the studies have been carried out with great care, yet before warranting publication in Cancers several points/issues should be addressed.
The conclusion of the article rightly points out that “subtle changes […] significantly influence the MDR-selective toxicity”. However, when looking at the structures of Q1-Q4 (active compounds) and NC-1 & NC-2 (non-chelating) or even NSC297366, significant differences appear that may render NC-1 and NC-2 not entirely relevant as control compounds.
Comparison between NSC297366 and Q4 can be straightforward (cyclic/open) but the data for NSC297366 are not included in this work. In this regard, NC-2 appears more like a control vs. NSC297366 than vs. Q-4. The relevance of NC-1 (a structural isomer) is less clear. A naphthol version of Q-3 or Q-2 (including the morpholine or piperidine moiety) would probably be more relevant as a non-coordinating control.
Eventually, a chlorinated Q-3 would also be interesting to allow comparison with Q-4 and assess more precisely the influence of Cl atom on the pKa of the OH and the activity.
The authors should synthetize some of these, evaluate their activity (lack of activity ?) and include these data in table 1.
Note: in Figure 1B the reacting naphthols should be swapped to match the final product of each reaction (in the supplementary the C-F couplings are missing from the 13C NMR description of NC-1 and NC-2).
Many experiments seem to have been carried out using the Cu or Fe complex rather than the ligand (see Figure 5 for example). Yet, the preparation or the structure of the complexes is not showed (one has to refer to ref 58, which only includes the structure of the Cu complexes). This should be included.
More importantly only the activity of the ligand is given (table 1). The authors assume that this activity is due to the in cellulo formation of the corresponding Cu or Fe complexes with endogenous metal so it is logical to assess the properties of theses complexes. However the cytotoxic properties of the free ligands and of the metal complexes can be different.
The authors should present both sets of results. This includes the toxicity of the complexes on MDR cell lines; but also the ROS production assay (Figure 5), which could be evaluated with the free ligands Q-1 to Q-4 as well to validate the conclusions regarding the involvement of ROS generation to explain the cytotoxicity of Q1-Q4. This would complement Figure 6 where the influence of NAC is shown on the activity of Q-1 to Q-4 (without the addition of metal salts).
The confusion between the activity of the “ligands” (Q) and of the complexes (“[QM]”) somewhat blurry the message of this study and makes it hard to draw clear conclusions, despite the interest of the topic.
Note: the abstract conclusion states: “Our results show that the MDR-selective anticancer activity of the studied 8-34 hydroxyquinoline derived Mannich bases is associated with the preferential formation of redox-35 active copper(II) complexes, which undergo intracellular redox-cycling to induce oxidative stress.” but the actual conclusion of the study is much more vague, which is a bit confusing (does that imply that the results are not 100% conclusive ?).
Minor comments:
Figure 6, to be consistent with Figure 2, maybe use pIC50 instead of IC50 ?
Typos: line 335 “dublicate”; line 472 “biologically”
Reviewer 3 Report
The authors present new insights on MDR describing a complementary method using various ligands and iron and copper chloride solutions.
The manuscript is well written but very hard to follow since most of the time they refer either to Suplimentary material, either to a previous work of the authors. I believe this can be improved in order to have the whole picture of what the authors already done and what this manuscript brings along as an original research. The results are valuable but not clear, for example the authors talk about NMR for the non-chelating compounds but no results there, also the cytotoxicity assays could be added in SA.
It is not clear how the results can be applied into preclinical practice, did the authors have in mind the develop an in vivo model using the cells line and test the toxicity.